# Boosting Fitness Costs Associated with Antibiotic Resistance in the Gut: On the Way to Biorestoration of Susceptible Populations

**DOI:** 10.3390/biom14010076

**Published:** 2024-01-08

**Authors:** Fernando Baquero, Jerónimo Rodríguez-Beltrán, Teresa M. Coque, Rosa del Campo

**Affiliations:** 1Department of Microbiology, Ramón y Cajal University Hospital, Ramón y Cajal Institute for Health Research (IRYCIS), 28034 Madrid, Spain; 2Network Center for Biomedical Research in Epidemiology and Public Health (CIBER-ESP), 28029 Madrid, Spain; 3Network Center for Biomedical Research in Infectious Diseases (CIBER-INFEC), 28034 Madrid, Spain

**Keywords:** fitness cost antibiotic resistance, bacterial stress in the gut, antibiotic susceptibility restoration

## Abstract

The acquisition and expression of antibiotic resistance implies changes in bacterial cell physiology, imposing fitness costs. Many human opportunistic pathogenic bacteria, such as those causing urinary tract or bloodstream infections, colonize the gut. In this opinionated review, we will examine the various types of stress that these bacteria might suffer during their intestinal stay. These stresses, and their compensatory responses, probably have a fitness cost, which might be additive to the cost of expressing antibiotic resistance. Such an effect could result in a disadvantage relative to antibiotic susceptible populations that might replace the resistant ones. The opinion proposed in this paper is that the effect of these combinations of fitness costs should be tested in antibiotic resistant bacteria with susceptible ones as controls. This testing might provide opportunities to increase the bacterial gut stress boosting physiological biomolecules or using dietary interventions. This approach to reduce the burden of antibiotic-resistant populations certainly must be answered empirically. In the end, the battle against antibiotic resistance should be won by antibiotic-susceptible organisms. Let us help them prevail.

## 1. Introduction

The acquisition of antibiotic resistance by horizontal gene transfer and by mutational changes in the chromosome or genes located in mobile genetic elements, followed by antibiotic resistance phenotype expression, implies modification of the cellular physiological status, including homeostatic adaptations of the previously susceptible cell [1]. Thus, in most cases, resistance imposes a type of stress, eventually resulting in a fitness cost, i.e., the growth rate of resistant bacteria might decrease. This cost is absent in susceptible bacteria; thus, hypothetically, the susceptible population is expected to prevail in the medium to long term. Unfortunately, the fitness cost of antibiotic resistance can be reduced after a certain period by the acquisition of compensatory mutations and possibly by phenotypic–epigenetic adaptations. In any case, such “secondary adaptations” can produce alternative rewiring of metabolic circuits, physiological deviations involving fitness costs. At their turn such new costs should be compensated, so that the long-term viability of resistant populations might be compromised.

## 2. Stress as a Relative Concept: The Case of Intestinal Microbiota

In nature, life is “struggle for life”; i.e., there is no utopia for microorganisms. Even those that are well adapted to their natural niches and that maintain their population in steady-state equilibrium fight against a multiplicity of stresses and do not grow as rapidly as those grown under optimal laboratory conditions (pure cultures, rich media, and controlled physical–chemical variables). Stress might be consequent to changes in the environmental physical–chemical conditions of the occupied niche, defined as a multidimensional environmental space characterized by a variety of conditions, both biotic and abiotic, whose quantitative ranges determine the positive or negative growth rates of the bacterial species [2,3]. Negative growth rates are frequently part of the stress phenotype. The altered-niche hypothesis as a source of stress for the occupant bacterial population can be extended to the whole ecosystem [4,5], including the entire microbiota. The intestine is a flowing open environment (an “invironment”) subject to the host’s circadian rhythms and perhaps influencing the microbiota over a 24 h cycle [6]. It could be true that the most host-adapted populations (the phyla Bacteroidota and Bacillota), with the highest densities in the normal human or most mammals’ microbiota, are generally those that constitute the older population in the co-evolutionary history of intestinal microbial colonization [7]. Thus, they are able to cope with ample ranges of changing conditions and are thereby subject to less stress. Abrupt changes in conditions (i.e., particular biomolecules or chemical conditions) might produce more stress than smooth changes, which frequently require simpler modifications. Most potential human pathogens that are frequently present in the human microbiota, such as the gamma-proteobacteria (Enterobacterales) *Escherichia coli*, *Klebsiella*, *Serratia*, or *Enterobacter*, constitute a very small proportion of the microbiota (on average less than 1%). Some of these, such as *E. coli*, are likely of more recent evolutionary acquisition in vertebrates, including mammals [8,9].

However, subpopulations of commensal opportunistic pathogens are frequently involved in urinary tract and bloodstream infections, producing outbreaks both in hospitals and community settings [10]. The proportion of these organisms (also *Enterococcus*) in the microbiota increases in aged and hospitalized individuals and in people from low-income countries with inadequate sanitation [11,12]. Consequently, the acquisition of antibiotic resistance traits by pathogenic clones of these species is of particular clinical relevance. Our hypothesis is that the fitness cost produced by the expression of antibiotic resistance in pathogenic organisms (mostly *Enterobacteriaceae*, or *Enterococcus*) might be increased by altering the surrounding eco-active intestinal chemosphere, resulting in relative fitness changes: the capability of a genotype or individual to survive and reproduce in comparison with a second genotype or individual [13]. This change could result in a disadvantage in relation to antibiotic susceptible populations that might replace the resistant ones. This outcome has been confirmed by fecal microbiota transplantation where antibiotic-resistant high-risk clones, were replaced by susceptible clones from the fecal donor [14,15,16]. A schema of the different stresses to which potentially pathogenic bacteria are exposed during their transit in the intestinal tract is presented in Figure 1.

## 3. The Main Sources of Bacterial Stress in the Intestinal Microbiota

### 3.1. Acid Stress

One of the first sources of stress faced by bacterial organisms after ingestion is gastric acidity. Under fasting conditions, hydrochloric acid in gastric juice has a highly aggressive pH of 2, whereas *Enterobacteriaceae* have an optimal pH in the neutral range, between 6.5 and 7.5 (pH 7 for *E. coli*). Intraluminal acidity is not exclusive to the stomach; the distal duodenum has an acid environment (pH 6), maintained by lactic acid bacteria, which increases until reaching an optimal pH of 7.4 in the terminal ileum, probably where most microbiota growth takes place. Again, acidity increases (probably because of the bacterial production of acids) to a pH of 5.7 in the cecum, ultimately reaching a pH of 6.7 in the rectum. Some pathological conditions might increase intestinal acidity, such as ileocecal resection, chronic pancreatitis, cystic fibrosis, ulcerative colitis, or Crohn’s disease [17]. This increase could be a direct consequence of the altered microbiota, which also occurs with the use of probiotics; in both cases, organic acids are involved. Organic short-chain carboxylic acids, such as fumaric, propionic, acetic, lactic, and butyric acids, are frequently produced by intestinal microorganisms, lowering the pH, which results in stress for a number of bacterial populations [18].

Adaptation to acid stress is an important factor for the transmission of intestinal microbes. In *E. coli*, resistance to acid stress is guaranteed by the GadE-regulated expression of glutamate and arginine decarboxylases associated with amino acid and Cl^−^/H^+^ antiporters [19,20]. Acid stress also forces *E. coli* to alter the envelope structure and porins in the outer membrane and the cytoplasmic chaperones [21]. Changes in the envelope architecture and in the molecular folding have epistatic consequences altering cellular functions [22]. In fact, there are several overlapping acid survival systems with variable expression and efficacy depending on the growth phase [23,24]. Resistance in *E. coli* to short-chain organic carboxylic acids also involves changes in the *rpoA* (influencing folding efficiency and/or chaperone-like activity), *rpoC* (subunit of RNA polymerase), and *rpoS* (alternative sigma factor inducing stationary phase) involved in stress response cascades, and probably *rho* (transcription regulation) and *nagA* (*N*-acetyl-d-glucosamine metabolism) [25,26]. As we discuss in the following paragraphs, in addition to the relatively low duodenal pH, this upper part of the small intestine has other sources of bacterial stress [27].

### 3.2. Bile Stress

Bile is stored in the gallbladder and flows into the duodenum by the common bile duct. Taurocholate, glycocholate, and glycochenodeoxycholate are the main bile salts, acting on bacterial membranes of several microorganisms and resulting in potent antimicrobial activity, mostly derived from the highly lipophilic steroid ring. Bile salts contribute to the host’s resistance to upper intestinal bacterial colonization. In the intestine, primary bile acids are susceptible to microbial-mediated oxidation, dihydroxylation, and epimerization, giving rise to the secondary bile acids deoxycholic and lithocholic acid. Bacterial stress derives from cell envelope stress, dissociation of integral membrane proteins, action on membrane lipids, alteration of nutrient uptake, reactive oxygen species-derived nucleic acid damage, and protein misfolding, eventually leading to a bactericidal effect [28,29]. In general, Bacillota, which include the opportunistic pathogen *Enterococcus faecalis* [30], are more sensitive to the deleterious effects of bile than Enterobacterales, given that the outer membrane’s lipopolysaccharide acts as a protection shield. However, there is also severe bacterial stress in this enteric group, inducing DNA damage, SOS gene stress, and hypermutation [31]. General stress proteins are expressed during *E. faecalis* bile salt treatment, including molecular chaperones and protectors of DNA-damaging peroxides [29]. Lastly, bilirubin excreted by the bile can have an antibacterial effect on Gram-negative bacteria [32].

Antibiotic-susceptible bacteria respond to the challenge of stress using bile efflux pumps for overexpression of MdtEF-TolC, particularly in acid medium, but at the expense of a fitness cost [33], bile salt hydrolase enzyme, and rewiring the intracellular metabolism and the cell membrane composition [34]. In *E. coli*, some mutations are bile-hypersensitive such as in AcrAB, EmrAB, and MdtABCD efflux pumps; in OmpF/OmpC outer membrane porin; in HupAB DNA-binding protein (involved in DNA supercoiling); and in genes biosynthesizing the core lipopolysaccharide, showing their effect on bile-resistance. The *E. coli* SOS gene, *dinF*, which protects against oxidative stress, also protects from the effect of bile salts [35]. The membrane damage sensors, Cpx and RcsCb, regulate and induce the expression of genes involved in bile stress responses [29].

### 3.3. Stress by Pancreatic Enzymes

Digestive enzymes such as amylase, lipase, trypsin, and chymotrypsin are released from the pancreatic acini cells (exocrine glands) and flow into the pancreatic duct to reach the duodenum. Lipases might have antibacterial activity, preferentially in Bacillota [36]. Trypsin and chymotrypsin, preferentially in combination, can hydrolyze bacterial outer-membrane proteins in Gram-negative organisms and damage the integrity of surface structures in Bacillota [37]. The extent of these effects in the large intestine is counteracted by trypsin degradation by commensal bacteria [38].

### 3.4. Stress by Short- and Long-Chain Fatty Acids

Intestinal short-chain fatty acids (2–6 carbons in length) are mostly produced by microorganisms acting on carbohydrates and polyphenols, and these compounds have significant effects in terms of reducing optimal bacterial fitness. Bacillota are mainly butyrate producers, whereas Bacteroidetes excrete acetate and propionate [39,40]. Although these effects are in part due to the reduction in pH (previously treated), they also have a pH-independent antibacterial mode of action. The long-chain fatty acids (12–20 carbons in length) present in the intestinal lumen originate from the host cells, the microbiota, and from dietary sources. The most abundant of these are unsaturated fatty acids, such as oleic and linoleic acids, and saturated fatty acids, such as stearic or palmitic acid. Free fatty acids are bound and further enzymatically released from other compounds, such as glycerol, sugars, or phosphate headgroups, to form lipids [41]. Their lipophilic nature allows them to invade and damage microbial membranes, ultimately leading to a lethal effect, particularly in Bacillota. Gram-negative bacteria such as Enterobacterales are protected in part due to their lipopolysaccharide layer in the outer membrane. However, Enterobacterales might be able to sense extracellular long-chain fatty acids by using a 2-component system that influences gene regulation. Consequently, general metabolism, type 3 secretion systems, or the gene network involved in motility, fimbriae synthesis, and biofilm formation, can be modified, influencing global bacterial fitness [42]. Such effects on gene expression are in part linked to the fact that fatty acids might mimic diffusible signal factors [43].

### 3.5. Stress by Dietary Compounds

Food–microbiota interaction is one of the cornerstones of intestinal physiology [44]. Among the roles of the microbiota in the assimilation of nutrients by herbivore animals is to degrade complex vegetal molecules (such as cellulose) by symbiotic cellulolytic bacteria, release oligosaccharides, and produce absorbable short-chain fatty acids, ensuring animal nutrition. To a minor degree, intestinal bacteria in humans (such as *Enterococcus*, frequent in the elderly) contribute to the degradation of complex polysaccharides [45]. However, food–microbiota interactions can result in a challenge for bacterial populations. Some of these causes of stress are examined below.

#### 3.5.1. Stress by Polyphenols

Polyphenols, complex natural molecules containing one or more hydroxylated aromatic rings, are a widely and highly distributed group of diverse natural products (probably over 10,000) found in dietary products such as fruit, vegetables, nuts, seeds, red wine, beer, olive oil, honey, coffee, and tea. Polyphenols, for instance flavonoids and tannins, have been shown to exert antibacterial effects, both in Bacillota species, such as *Staphylococcus aureus*, and in gamma-Proteobacteria, such as *E. coli*, *Klebsiella pneumoniae*, *Acinetobacter*, and *Pseudomonas* [46]. In addition, many of them have synergistic activity with antimicrobial agents [47,48]. The antibacterial mode of action of flavonoids appears to involve the perforation and destructuration of the bacterial cytoplasmic membrane, alteration of bacterial transporters, DNA topoisomerase inhibition, and reduction of bacterial energy metabolism by inhibition of nicotinamide adenine dinucleotide hydrogen reductase, all of which are various mechanisms that result in the formation of lethal reactive oxygen species [49,50,51,52]. Bacterial resistance to polyphenols is a poorly explored field of research; however, microorganisms can produce degraded polyphenols, activating glycosidases and esterases, isomerases, and hydrolases, giving rise to simple aromatic metabolites [51]. To which extent these activities are induced by polyphenol stress remains poorly understood.

#### 3.5.2. Stress by Polyamines

Decarboxylation by intestinal microorganisms (mostly anaerobes, such as *Bacteroides* or *Fusobacterium*) of aromatic or polycationic amino acids results in polyaminated molecules, biogenic amines, and polyamines. Polyamines include compounds with two amino groups, such as putrescine (1,4-diaminobutane) or cadaverine (1,5-diaminopentane), but also molecules with three or four amino groups, such as spermidine [N-(3-aminopropyl)butane-1,4-diamine] and spermine [N,N′-bis(3-aminopropyl)butane-1,4-diamine], respectively. Bacteria have transport systems allowing uptake of extracellular polyamines, including the polyamine ABC transporter genes, generally organized as four-gene operons, as in the cases of *pot*ABCD (spermidine uptake) and *pot*FGHI (putrescine uptake). These compounds have long been known as antibacterials [53], acting on Bacillota species and on those of the family Enterobacteriaceae. They alter bacterial membrane permeability and porin function, they possibly interact with nucleic acids, and these effects are likely highly concentration-dependent. In any case, they have been considered to constitute possible scaffolds for novel antimicrobials or antibiotic enhancers [54,55]. Possible mechanisms of resistance to polyamines involve mutations in these genes or downregulation of operon transcription. However, polyamines might also provide benefits for the bacteria, providing, e.g., resistance to acidity or protection against oxidative stress [56].

#### 3.5.3. Nitric Oxide Stress, Osmolar Stress

Dietary nitrates and nitrites are widespread in food, and they are found naturally in vegetables and fruit or as food additives. They give rise in the gut to reactive nitrogen species and to a human intestinal inflammatory response. Occasionally, bacteria lead to an overproduction of nitric oxide in the gut, with potential antibacterial activity based on lipid peroxidation, nitrosation of membrane proteins, and DNA damage [57,58,59]. The cellular targets of nitric oxide and reactive nitrogen species act as signals, resulting in altered gene expression and synthesis of protective detoxifying enzymes [60]. Osmolarity essentially influences bacteria during their flow or during transient colonization of the small intestine and depends on unabsorbed meal compounds. Osmolality, the concentration of solute particles in a solution, also influences bacterial populations. There is a reduction in bacterial cell volume due to passive water excretion [61]. Bacteria adapt to osmolarity stress by accumulating solutes, such as potassium, glutamate, trehalose, proline, and glycine betaine [62].

### 3.6. Stress by Nutritional Deficiency

Accessible nutrients for the microbiota in the intestine are always limited, for three main reasons: (1) the host and microbiota compete for nutrients, so that only a small part of dietary food is available for the microbiota; (2) the great density of bacterial cells in the most colonized, anaerobic, and dehydrated part of the intestine, the colon, leads to inter-microbial competition for nutrients; and (3) microbial populations lost daily by defecation need replacement, so doubling time in the gut by a day or more would not be sufficient to maintain a stable population size. It had been proposed that bacterial abundance in the gut fluctuates around the stable carrying capacities of the colonizable gut [63]; thus, many bacterial populations are challenged by conditions close to starvation [64]. It should be noted that nutritional conditions vary along the intestine, being more favorable around the ileocecal valve and proximal colon, probably making it the most effective “growth zone” [65]. Bacterial nutrients from the ileum are dietary but undigested fiber polysaccharides, and secondarily host mucosal glycans and host secretions, as well as microbial exopolysaccharides and capsular material [66]. In the colon, extreme interbacterial competition for nutrients, including nitrogenated compounds and vital metals such as iron or even vitamins, also absorbed by the host, overcomes the presumed higher concentration of these nutrients by host water absorption (also deleterious substances for bacteria concentrate, increasing toxicity) and intermicrobial nutritional cooperation. Microbes in the gut have access to only 1 nitrogen atom for every 10 carbon atoms, whereas free-living organisms (let alone cultures in the lab) have access to 4 nitrogen atoms for every 4 carbon atoms [67]. Bacterial reactions to nitrogen starvation stress in *E. coli* include global physiological changes (stringent response) mediated by the signal molecule, guanosine tetraphosphate (ppGpp) [68]. In general, nutrient starvation, including inorganic phosphate starvation, produces similar responses, leading to bacteria entering a stationary phase [69]. A poorly explored point is how microbial nutritional starvation influences majority and minority gut populations. The populations with higher densities are probably more resilient to extinction, given that “number is a biological advantage”, as occurs under antibiotic exposure [70]. Many antibiotic-resistant pathogenic bacteria are minorities (less than 1% of the population), and in the absence of antibiotic exposure, resistant populations within a species are also minorities. However, their number generally increases in hospitalized patients.

### 3.7. Stress Resulting from Microbial Interactions

#### 3.7.1. Stress by Bacterial Antimicrobial Peptides: Microcins, Lantibiotics, Colicins

Microcins are low-molecular-weight antibiotic peptides. They were distinguished in 1976 from colicins, which are higher-molecular-weight antibacterial proteins that are much less stable in the intestinal tract [71]. Microcins are ribosomally synthesized and post-translationally modified peptides (RiPPs), which are mostly produced by Enterobacteriaceae and act on members of this family of microorganisms. They have various mechanisms of action, such as producing pores in the cytoplasmic membrane (MccV, MccE492, and MccL); inhibiting the aspartyl-tRNA involved in protein synthesis (MccC), inhibiting the topoisomerase GyrB, producing double DNA breaks (MccB17); blocking the secondary RNA polymerase channel, impairing transcription and acting on cytochromes inhibiting cellular respiration (MccJ25); or altering the function of the cellular proton channel (MccH47, and possibly MccM and MccI), or the ATP synthase (MccH47). Microcin stress is followed by immunity/resistance mechanisms, including acetyltransferases (MccC), production of immunity proteins (Class IIb microcins), enhanced efflux pumps, and inhibition of DNA gyrase supercoiling activity (MccB17). There is evidence that microcins strongly influence microbial interactions in the gut [13]. The equivalent of microcins in Enterobacteriaceae are lantibiotics in Bacillota, as well as ribosomally produced and modified post-translational peptides [72]. Lantibiotics (lanthionine- and methyllanthionine-containing peptides) can produce holes in the bacterial membrane and eventually interfere with cell wall synthesis [73]. Resistance/protection from lantibiotics is mediated by the production of “immunity proteins”, specialized ABC-transport proteins, modifications in membrane composition, lantibiotic–lytic proteins, spore formation, and immune mimicry [74]. Colicins are much larger and less stable polypeptides in the intestinal environment, and they are produced and active in Enterobacteriaceae. Their mechanism of antibacterial action includes membrane pore formation, degradation of nucleic activity (DNase, 16S rRNase, and tRNase activities), and altering peptidoglycan synthesis. Resistance to colicins involves receptors and translocation mutants (Tol pathway mutants), alteration of outer membrane proteins, including *ompF*, *exbB*, and *tonB* mutations, and enterochelin hyperproduction [75].

#### 3.7.2. Stress by Bacteriophages and Microbial Predators

Bacterial viruses (bacteriophages) have been postulated to be the most abundant microorganisms in the gut. However, most of these phages are prophages, or lysogenic phages that replicate with the host bacterial strain. The “free” (extracellular) phages, which are able to infect new organisms, are comparatively smaller in abundance but can locally increase in number and evolve into a bacteriolytic state when induced (activated) by stressful conditions [76]. The most abundant viral families include *Myoviridae*, *Podoviridae*, *Siphoviridae*, and *Microviridae*. Most bacterial stress produced by phage invasions derives from envelope (cytoplasmic membrane) stress, fostering a phage-shock-protein (Psp) system, occurring both in Gram-positive and Gram-negative microbes [77]. Classic mechanisms of resistance to phage invasions are alterations in bacterial surface epitopes acting as phage receptors and restriction-modification systems. There is also the production of proteins interfering with the phage infection cycle, and these include variable and evolving CRISPR sequences. There are also phenotypic mechanisms of resistance, which change the metabolic status of the cell and are similar to antibiotic persistence in bacteria [78]. Comparative stress by bacterial predators, such as protozoa, appears to have less importance. However, commensal protozoa, such as *Entamoeba* or *Blastocystis*, eating bacteria, might be important for microbiome stability in low-income human populations, particularly in the proximal gut [79]. Bacterial resistance to protozoa is analogous to resistance to phagocytosis and survival in phagolysosomes [80]. The classically described environmental predators, such as *Bdellovibrio*, can also be abundant in the gut. they penetrate the cell and multiply in the periplasm, killing the prey bacterium; the process is probably too rapid to produce a significant population of stressed bacteria [81].

### 3.8. Stress by Inflammation and Immunity

Frequently, the relationship between microbiota and the host (particularly in mucus-associated bacteria) can present as a status of “low-grade inflammation”, mostly induced by bacterial exopolysaccharides and cell wall fragments. Innate immune defense is exerted by the secretion of specialized epithelial cells (Paneth cells) of antimicrobial peptides such as α-defensins, which interact and disorganize bacterial membranes, eventually resulting in cell death [82,83]. These cells also produce other antimicrobial peptides, such as CRS4C and the lectin Reg3γ, which disrupt the cell wall [84]. Also, cathelicidins (including indolicidin), produced by intestinal epithelial cells, have significant antibacterial activity [85,86]. If the secretion of these antimicrobial peptides is constitutive, the local invasion by microorganisms could increase their concentration, so that invasive antibiotic-resistant pathogens are facing a higher stress. Mechanisms of bacterial resistance might evolve in Enterobacterales by alteration of the outer membrane lipopolysaccharide. Other molecules of the immune system, such as the complement system, are probably involved in the stress of bacterial cells in contact with the epithelium [87]. Toll-like host receptors recognize microbial-associated molecular patterns, and enterocytes express various complement components. Complement proteins are found among the bacterial-bound proteins detected in intestinal proteomic studies (Concepción Gil, personal communication), and they might kill bacteria directly via large pore-forming complexes [88]. Bacterial resistance to defensins is mediated by expressing proteins such as MprF, which harbors transmembrane domains for lipid lysinilation and defensin repulsion [89,90]. Perhaps as a consequence, MprF plays a crucial role in *Staphylococcus aureus* virulence and is involved in resistance to daptomycin, which is structurally similar to cationic antimicrobial peptides. Similar effects occur in *Enterococcus faecium* [91,92].

## 4. Modulating Intestinal Stress to Select for Antibiotic Susceptibility

The main purpose of this review was to examine the possibility that regulating/modulating or administering physiological molecules of the intestinal tract, which enhance gut stress, might result in fitness costs on microorganisms invading or colonizing the gut (Figure 2).

If the expression of resistance mechanisms, including the carriage of mobile resistance elements, in the absence of antibiotics, challenges the bacterial physiology and produces a fitness cost [93,94], it could be of interest to know whether, in the absence of antibiotic exposure, the addition of both types of fitness cost could be untenable for antibiotic- resistant populations but not for susceptible ones. This hypothesis, which suggests that stressful conditions in the gut could exacerbate the fitness costs of resistance, is substantiated by various lines of evidence. Although the precise mechanisms underlying the fitness cost of resistance remain largely unknown, the physiological changes induced by expressing antimicrobial resistance genes overlap with those caused by previously mentioned stressors.

For instance, beta-lactamase expression produces a fitness cost due to the accumulation of enzymes in the periplasmic space, destabilizing the bacterial envelope [22,95,96]. Similarly, bacteriocins, long-chain fatty acids, bile salts, and dietary by-products (among other stressors) exert antimicrobial activity by damaging the cell’s envelope. Therefore, the physiological impact of expressing a β-lactamase and undergoing gut-associated stress will likely synergize, resulting in an unsustainable fitness cost for the antibiotic-resistant bacteria. A different situation emerges if gut stressors enhance antibiotic effectiveness, which is particularly plausible for antibiotics targeting the bacterial envelope (e.g., β-lactams and polymyxins) or those relying on reactive oxygen species for their killing mechanism (e.g., aminoglycosides and quinolones). As mentioned earlier, harsh gut conditions often impact the bacterial envelope and generate oxidative species (such as nitric oxide or bile salts), which could lead to a synergistic effect with specific antibiotics, although further demonstration of this interaction is needed.

Two aspects should be clearly differentiated. First, there is the interaction of intestinal stress with the antibiotic-provoked stress. Some studies have suggested that normal mechanisms of bacterial decontamination in the gut (such as bile production) could increase the antibacterial effect of antimicrobial drugs [97]; furthermore, mechanisms of resistance to gut antibacterial products/conditions could favor cross-resistance with antibiotics [98]. We cannot discard pleiotropic fitness costs, meaning the mechanisms of resistance to gut physiological conditions might result in higher antibiotic susceptibility. Nor can we rule out the possibility that antibiotic resistance could reduce the possibilities of gut invasion or colonization, following source –sink dynamics [99]. Second, there is the interaction of intestinal stress with the fitness associated with antibiotic resistance. The coincidence of two or more stresses might not only have a synergistic activity, pushing populations toward extinction, but could also result in a reduction in mutational or phenotypical adaptation, particularly if cases of antagonistic pleiotropy (collateral susceptibility) could be demonstrated. For instance, bile salts and sodium deoxycholate are more active against erythromycin-resistant *Campylobacter coli* strains than against erythromycin-sensitive strains [100]. Incoming antibiotic-resistant microorganisms into the gut might also have been “previously stressed” in processed drinks or food [101]. We cannot rule out the possibility of unwanted effects if two different types of stress could produce less effect on the fitness cost than a single one. Unfortunately, the effects of merging intestinal environmental stress and antibiotic stress in susceptible and resistant bacteria have scarcely been explored. These studies could help to pharmacologically modulate the intestinal biomolecules or particular biological effectors to favor antibiotic-susceptible populations. Fitness costs associated with the expression of antibiotic resistance mechanisms and/or with the carriage of mobile genetic elements could be unbearable for certain resistant bacterial populations, favoring their replacement with the susceptible ones. Stated another way, given that life is a nonequilibrium phenomenon, we propose to act against resistance by modifying energetic flux balances [102], thus altering the relative fitness costs of susceptible and resistant populations. As shown in Figure 1, the supply of energy (ATP-producing processes) has a cost, which is balanced by the benefits of energy investment (ATP-consuming processes), resulting in bacterial replication. Everything is a balance of costs and benefits, leading to sinks and sources.

## 5. Boosting Fitness Costs of Antibiotic-Resistant Organisms in the Gut: A Testable Hypothesis

Lastly, we conclude that we are proposing a testable hypothesis. Fitness costs associated with bacterial intestinal stress might differ in antibiotic-susceptible and antibiotic-resistant populations of bacterial pathogens; however, the current available information is extremely scant. To calculate the relative fitness of both resistant and susceptible populations, we can approach high-throughput competition assays using flow cytometry, as previously described [103]. Antibiotic-susceptible and -resistant variants (mutations and resistance plasmids) are introduced in the same (isogenic) strain carrying a plasmid containing a green fluorescent protein (*gfp*) gene inducible by arabinose. These co-cultures could be exposed (several replicates) to various concentrations of intestinal stress molecules and conditions, with appropriate controls to calculate the fitness cost of resistance without these ecological stresses. As a confirmatory experimentation, animal models can be used in competition experiments based on oral inoculation with pairs of isogenic susceptible and resistant strains at the same cell density, to ascertain the competitive advantage of susceptible ones. Differences in the relative fitness of resistant and susceptible populations might suggest a list of gut physiological molecules to be targeted. This can be followed by the development of natural products or drugs boosting stress molecules for resistant bacteria. Nutritional interventions are also possible [104], as components of the diet might favor colonization by resistant bacteria [105,106]. Moreover, a diet assuring microbial diversity also protects against incoming resistant organisms [107]. If interventions directed to enhance to fitness costs associated with antibiotic resistance might influence the survival of commensal organisms populations expressing and spreading resistance, including anaerobes [108] is also an interesting possibility.

These approaches to reducing the burden of antibiotic-resistant populations can be and certainly have to be answered empirically. In the end, the battle against antibiotic resistance should be won by antibiotic-susceptible organisms. Let us help them prevail.

## Figures and Tables

**Figure 1 biomolecules-14-00076-f001:**
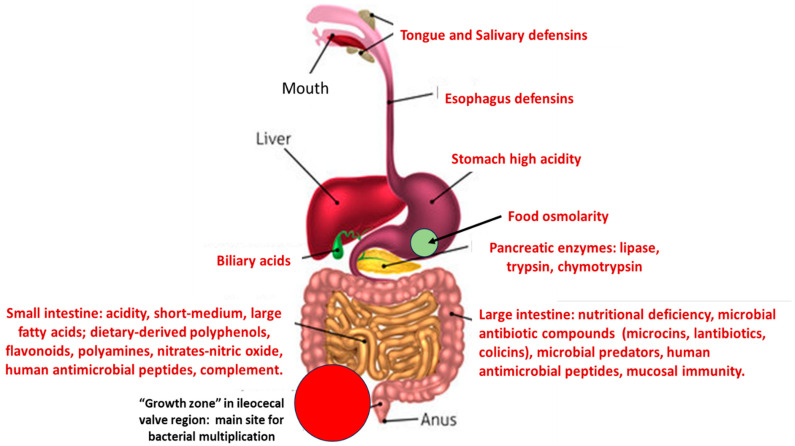
A schema of the main stresses to which potentially pathogenic/antibiotic resistant bacteria are confronted in the human gut, producing a reduction in fitness.

**Figure 2 biomolecules-14-00076-f002:**
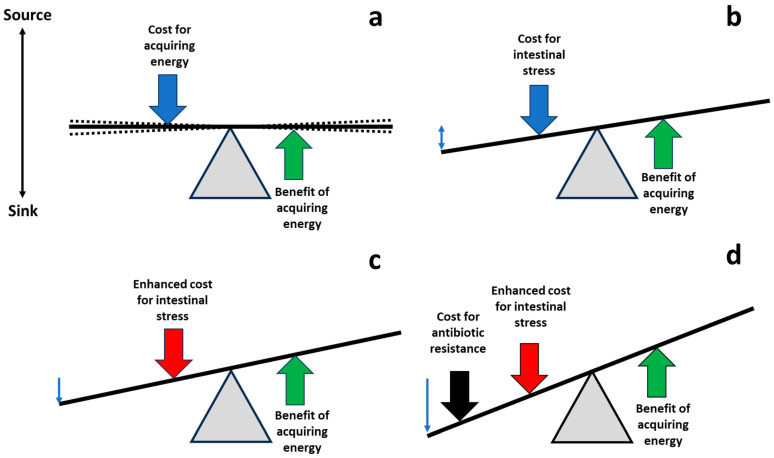
Shifting the balance of costs and benefits to favor antibiotic-susceptible populations. (**a**) Life is based on equilibrium (with oscillations, broken lines), balancing the costs of acquisition of energy required for fueling biological processes (blue arrow), and the benefits of energy investment, leading to the final goal of bacterial replication (green arrow). (**b**) During the process of gut invasion and colonization, potentially pathogenic/resistant bacterial populations are exposed to intestinal molecules, reducing their bacterial fitness; compensatory adaptations also contribute to this fitness cost (blue arrow). (**c**) Managing pharmacological physiological molecules in the gut, it could be possible to increase the cost of these populations in the intestine (red vertical arrow). (**d**) The increase in intestinal fitness cost might be additive or synergistic with the fitness cost associated with the expression of antibiotic resistance or the cost of harboring carriers (mobile genetic elements) of antibiotic resistance (black arrow). Thus, the antibiotic-susceptible intestinal populations of potential pathogens could have better fitness than the resistant ones, favoring a restoration of susceptibility.

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
