# Peer review of "Boosting Fitness Costs Associated with Antibiotic Resistance in the Gut: On the Way to Biorestoration of Susceptible Populations"

_biomolecules, 2024, doi:10.3390/biom14010076_

Round 1

Reviewer 1 Report

Comments and Suggestions for Authors

In this study, Baquero et al. reviewed bacterial stresses in the gut environment and how they increase the fitness cost associated with antibiotic resistance mechanisms. They hypothesized that the effect of various combinations of fitness costs should be tested in antibiotic resistant bacteria with susceptible ones as controls, which can provide opportunities to increase the bacterial gut stress using physiological biomolecules or derivatives of them. The authors first described the stress in detail, then the main sources of bacterial stress such as acid, bile, pancreatic enzymes, fatty acids, dietary compounds, polyphenols, polyamines, nitric oxide, nutritional deficiency, microbial interactions, inflammation, and immunity in the intestinal microbiota were explained. Each section is written in very detail and discussion of each one in a review paper is extremely important for the field.  Fitness costs of antibiotic resistant organisms in the gut is a well-known hot topic but the literature is extremely poor about the mechanisms behind of it. The review paper is not only explaining the stress conditions and fitness cost but also it proposes a testable hypothesis. Differences in the relative fitness of resistant and susceptible populations might suggest a list of gut physiological substances to be tested following the protocols for drug development, or eventually guiding interventions based in changes in human diets. The review is opening the door to aiding the battle against antibiotic resistance among antibiotic-susceptible organisms. The paper is exceptionally well-organized and written by experts in the field. Only minor English revisions are required before publication

Author Response

REVIEWER 1.

 Q1. In this study, Baquero et al. reviewed bacterial stresses in the gut environment and how they increase the fitness cost associated with antibiotic resistance mechanisms. They hypothesized that the effect of various combinations of fitness costs should be tested in antibiotic resistant bacteria with susceptible ones as controls, which can provide opportunities to increase the bacterial gut stress using physiological biomolecules or derivatives of them. The authors first described the stress in detail, then the main sources of bacterial stress such as acid, bile, pancreatic enzymes, fatty acids, dietary compounds, polyphenols, polyamines, nitric oxide, nutritional deficiency, microbial interactions, inflammation, and immunity in the intestinal microbiota were explained. Each section is written in very detail and discussion of each one in a review paper is extremely important for the field.  Fitness costs of antibiotic resistant organisms in the gut is a well-known hot topic but the literature is extremely poor about the mechanisms behind of it. The review paper is not only explaining the stress conditions and fitness cost but also it proposes a testable hypothesis. Differences in the relative fitness of resistant and susceptible populations might suggest a list of gut physiological substances to be tested following the protocols for drug development, or eventually guiding interventions based in changes in human diets. The review is opening the door to aiding the battle against antibiotic resistance among antibiotic-susceptible organisms. The paper is exceptionally well-organized and written by experts in the field. Only minor English revisions are required before publication.

A1. We thank this reviewer for their kind words and understanding of our main message. The manuscript has been slightly modified and we added some new references.

Reviewer 2 Report

Comments and Suggestions for Authors

Antimicrobial resistance (AMR) is a growing global crisis which already kills millions of people each year. In the face of this, there is increasing interest in developing interventions that could replace antibiotics, including much scrutiny of so-called “probiotic” efforts in which existing normal flora could be boosted, or exogenous normal flora added, to counteract pathogenic and/or antibiotic resistant microorganisms. This well-written article from Baquero et al. is a hybrid sort of article type that has been submitted as a review, but in fact is largely concerned with proposing a testable hypothesis: that stress in the gut might be leveraged to oust AMR organisms amidst a milieu of competing sensitive organisms. The review component is comprehensive and underscores the immense complexity of the ecosystem – which ironically renders the final proposed experiment to test the hypothesis in vitro as too massively simplified to be relevant. Still, it is thought-provoking and may lead to new areas of inquiry.

Some specific comments follow: 

1.     This is not strictly speaking a review article. It is down to the editor to decide if the article type designation is correct and whether this conforms to the aims and scope.

2.     The authors make no reference to antibiotic phenotypic resilience behaviors that are downstream of DNA – eg tolerance, persistence. How do these phenomena play into the scenarios if at all?

3.     Lines 35-38. Penultimate sentence suggests that these "secondary adaptations" result in a reduced fitness cost in resistant populations due to acquired compensatory mutations. The last sentence then suggests that these secondary adaptations can compromise the viability of resistant populations. I assume this is meant to demonstrate the ‘cat and mouse’ nature of mutations and counter-mutations, but could the authors make this a bit clearer? (also, “at their turn” might be clearer as “in turn”, which implies responsive countermeasure).

4.     Lines 50-51 – circadian rhythms are just one of a bewildering array of complexities and it seemed arbitrary to single them out without good reason.

5.     Sentence in 56-58 (rough vs smooth consequences): what is the evidence for this statement (reference[s])?

6.     Sentence 61-62 (mammalian evolution): what is the evidence for this statement (reference [s])?

7.     Sentence 64-66 (age/hospitalization/LIC): what is the evidence for this statement (reference [s])?

8.     Line 68-72 – Perhaps I am just missing something, but this seems to imply the AMR element is chromosomal, when it is frequently on an MGE that can be flexibly lost as needed. In this respect, surely having the stressors around will force the AMR gene to be lost, and this could more simply (Occam’s Razor) account for any improvements seen if stress is boosted?

9.     Line 77. The author’s unpublished data does not seem to me to “prove” the hypothesis as there might be other explanations. Furthermore I feel it is not a good idea to rely on such unpublished data to make the argument as we cannot see the data or know if it would pass peer review. It perhaps could be mentioned much more speculatively in the concluding discussion area. There are a number of published accounts of FMT – should these be mentioned?

10.  Line 112 – interesting but would benefit from a concluding sentence or two to suggest how it would support the hypothesis.

11.  Figure 2. I simply could not understand this nor the text arguments supporting it. Perhaps re-write and run it past someone with little knowledge of such theory. It would reach a broader audience if this could be explained better to a non-specialist who has come to the paper from other areas (clinicians, microbiologists etc).

12.  Line 407 made me wonder if there was any room for a systems biology/mathematical modelling approach to making sense of things amid all this complexity? Has this been attempted in the field? What were the key findings?

13.  In the end I felt what was missing from the conclusion was a sense of how each class of stress might relate to the hypothesis. Also missing would be how one might start to approach specific interventions. Has anyone ever tried it even if they weren’t aware of the hypothesis? 

14.  A schematic figure for the key experiment proposed from line 418 would be useful.

Author Response

REVIEWER 2.

Q1. Antimicrobial resistance (AMR) is a growing global crisis which already kills millions of people each year. In the face of this, there is increasing interest in developing interventions that could replace antibiotics, including much scrutiny of so-called “probiotic” efforts in which existing normal flora could be boosted, or exogenous normal flora added, to counteract pathogenic and/or antibiotic resistant microorganisms. This well-written article from Baquero et al. is a hybrid sort of article type that has been submitted as a review, but in fact is largely concerned with proposing a testable hypothesis: that stress in the gut might be leveraged to oust AMR organisms amidst a milieu of competing sensitive organisms. The review component is comprehensive and underscores the immense complexity of the ecosystem – which ironically renders the final proposed experiment to test the hypothesis in vitro as too massively simplified to be relevant. Still, it is thought-provoking and may lead to new areas of inquiry.

A1. We are very grateful to this referee for his/her understanding of our main concept. We certainly agree with the feeling that everything in this field is massively complex, but that is the horizon of biological sciences in our days. We just propose to start experimenting the hypothesis, using competition experiments, first in vitro and then in experimental animals. These experiments could include in a next step some type of intervention by using physiological enhancers of the bacterial stressful factors. This might lead to “news areas of inquiry”.

 Some specific comments follow: 

Q2.  This is not strictly speaking a review article. It is down to the editor to decide if the article type designation is correct and whether this conforms to the aims and scope. 

A2. This is a “creative review article”, in the sense that uses available published information to orientate a new field of research. We understand that we need a synthetic dimension of scattered pieces of knowledge to  facilitate the advancement of science.

Q3. The authors make no reference to antibiotic phenotypic resilience behaviors that are downstream of DNA – eg.  tolerance, persistence. How do these phenomena play into the scenarios if at all? 

A3. The main message of our paper is if the passage of resistant bacteria through the gut implies increases in fitness costs that could be added to the costs of expressing resistance. I agree that persistence (phenotypic resistance) could also be a result of gut stresses, and that is mentioned in the text. However, that will produce (as tolerance) an antibiotic-resistant phenotype, and therefore the cell do not need to express antibiotic resistance mechanisms to survive. Even if they were expressed (I cannot find experiments about that in the literature) they will probably not selected, as the cells expressing them will not have advantage in organisms that are resistant persisters. Therefore, I do not expect “fitness cost additivity” in persisters or tolerant cells.

Q4. Lines 35-38. Penultimate sentence suggests that these "secondary adaptations" result in a reduced fitness cost in resistant populations due to acquired compensatory mutations. The last sentence then suggests that these secondary adaptations can compromise the viability of resistant populations. I assume this is meant to demonstrate the ‘cat and mouse’ nature of mutations and counter-mutations, but could the authors make this a bit clearer? (also, “at their turn” might be clearer as “in turn”, which implies responsive countermeasure).

A4. Thanks, I agree there was a need of clarification in this paragraph, which was modified in wording and structure in the revised text.

Q5. Lines 50-51 – circadian rhythms are just one of a bewildering array of complexities and it seemed arbitrary to single them out without good reason.

A5. This was added to emphasize the constant changes in the ecology of gut microbiota, event along the day,  resulting in a constant adaptive stress. I added a few words to clarify the point.

Q6.  Sentence in 56-58 (rough vs smooth consequences): what is the evidence for this statement (reference[s])?

A6. This is an interesting point. Eco-evolutionary wisdom support that the demand for evolutionary shifts caused by rapidly changing environmental conditions might be associated to high fitness costs, that might lead even to extinction (Kristensen, T. N., Ketola, T., & Kronholm, I. (2020). Adaptation to environmental stress at different timescales. Annals of the New York Academy of Sciences, 1476(1), 5-22). The reason is that abrupt changes generally requires a complex (potentially disturbing) response, and adaptation to smooth environmental changes are solved with simpler bacterial changes. Bacterial adaptation to antibiotics benefits in fact from environmental gradients (Baquero, F.,  Coque, T. M. (2014). Widening the spaces of selection: evolution along sublethal antimicrobial gradients. MBio, 5(6), 10-1128)    However, I do not think that this clarification requires a particular reference. In any case, I added a few words to the sentence to clarify the matter.

Q7.     Sentence 61-62 (mammalian evolution): what is the evidence for this statement (reference [s])?

A7.In terms of evolutionary time (years), there are not much information.  However, wee have previously published phylogenetic reconstructions suggesting that there are  “recent” E. coli phylogroups, particularly involving phylogroup B2 (including important antimicrobial resistant clades, as E. coli ST131) , whereas other phylogroups, as A or B1 are closer to the “last common ancestor” of E.coli (Gonzalez-Alba, J. M., Baquero, F., Cantón, R., Galán, J. C. 2019. Stratified reconstruction of ancestral Escherichia coli diversification. BMC genomics, 20, 1-15). Consistently, phylogroup B1 is  more frequently present in fishes, amphibians, reptiles, or birds, whereas, phylogroup B2 is more frequently  present in “more recent” herbivore and omnivore vertebrates, including humans (Gordon, D. M., & Cowling, A. 2003. The distribution and genetic structure of Escherichia coli in Australian vertebrates: host and geographic effects. Microbiology, 149(12), 3575-3586). In the new version of the manuscript we have added these two references.

Q8.     Sentence 64-66 (age/hospitalization/LIC): what is the evidence for this statement (reference [s])?

A8. We added three illustrative references:

 Q9.     Line 68-72 – Perhaps I am just missing something, but this seems to imply the AMR element is chromosomal, when it is frequently on an MGE that can be flexibly lost as needed. In this respect, surely having the stressors around will force the AMR gene to be lost, and this could more simply (Occam’s Razor) account for any improvements seen if stress is boosted?

A9. You are right. The expression of AMR located in MGE generally produce more cost than in the chromosome (Vogwill, T., & MacLean, R. C. (2015). The genetic basis of the fitness costs of antimicrobial resistance: a meta‐analysis approach. Evolutionary applications, 8(3), 284-295), but plasmid loss is frequently associated with the extinction of the carrying clone, because of toxin-antitoxin systems. Therefore the elimination of AMR plasmids do not “restore susceptibility” in this lineage, as the clone is dead. Ideally, the best is “the loss of the AMR gene”, not the resistant clone (who might be “physiological” in the microbiome.  However, given the absence of data, we consider that this point should not be included in the text

 Q10.     Line 77. The author’s unpublished data does not seem to me to “prove” the hypothesis as there might be other explanations. Furthermore I feel it is not a good idea to rely on such unpublished data to make the argument as we cannot see the data or know if it would pass peer review. It perhaps could be mentioned much more speculatively in the concluding discussion area. There are a number of published accounts of FMT – should these be mentioned?

A10. You are entirely right. The statement is now more appropriately supported with three new references.

 Q11.  Line 112 – interesting but would benefit from a concluding sentence or two to suggest how it would support the hypothesis.

A11. I added a short sentence and a reference (already existing in the reference’s list, but in other place) to clarify the importance of changes in cellular envelope architecture and molecular folding in alterations of cell function.

Q12.  Figure 2. I simply could not understand this nor the text arguments supporting it. Perhaps re-write and run it past someone with little knowledge of such theory. It would reach a broader audience if this could be explained better to a non-specialist who has come to the paper from other areas (clinicians, microbiologists etc).

A12. The text of the Figure 2 was simplified and rephrased, probably improving clarity.

 Q13.  Line 407 made me wonder if there was any room for a systems biology/mathematical modelling approach to making sense of things amid all this complexity? Has this been attempted in the field? What were the key findings?

A13. We have been working along the last years in computational modeling of antibiotic resistance. See for instance: Campos, M., Capilla, R., Naya, F., Futami, R., Coque, T., Moya, A., ...  Baquero, F. (2019). Simulating multilevel dynamics of antimicrobial resistance in a membrane computing  model. MBio10(1), 10-1128; Campos, M., San Millán, Á., Sempere, J. M., Lanza, V. F., Coque, T. M., Llorens, C.,  Baquero, F. (2020). Simulating the influence of conjugative-plasmid kinetic values on the multilevel dynamics of antimicrobial resistance in a membrane computing model. Antimicrobial agents and chemotherapy64(8), 10-1128. I am confident that we can apply membrane-computational models to take advantage of the data that we propose to obtain with the methodology proposed in this hypothesis review.

 Q14.  In the end I felt what was missing from the conclusion was a sense of how each class of stress might relate to the hypothesis. Also missing would be how one might start to approach specific interventions. Has anyone ever tried it even if they weren’t aware of the hypothesis? 

A14. I am also missing the answers to these questions, as to my knowledge the experiments have not been done. In any case, I have expanded this paragraph, suggesting competition experiments in animal models with isogenic susceptible/resistant bacteria, and the possible role of nutritional interventions.

Q15.  A schematic figure for the key experiment proposed from line 418 would be useful.

A15. I agree, but we prefer to wait for solid experimental results in a next publication. In any case, the type of experiments to be performed have been now explained in more detailed way.

Reviewer 3 Report

Comments and Suggestions for Authors

The authors present a review of potential physiological stresses and propose to utilize these stresses and test the hypothesis that such stresses may affect the cost-benefit balance of drug-resistant opportunistic pathogens in the gut. This is a very important topic and will be of interest to a wide audience in the scientific community. The authors did an excellent job summarizing those physiological stresses and enhance their report with some schematics.

The authors do not address the issue of resistance gene transfer between non-pathogenic commensals and pathogens. As an example, the vanB-type vancomycin resistance transposon, which is commonly carried by anaerobic gut commensals of the phylum Firmicutes [Stinear TP, Olden DC, Johnson PD, Davies JK, Grayson ML. Enterococcal vanB resistance locus in anaerobic bacteria in human faeces. Lancet. 2001. Mar;357(9259):855–6. 10.1016/S0140-6736(00)04206-9]. Another issue is the difficulty of testing specific immunological stresses on gut microbiota or in the testing of the proposal provided by the authors. The review could also benefit the reader by considering the case of drug-resistant C. difficile. A real problem to address and a pathogen can apparently withstand the physiological stresses in the gut and is able to persist in the host and maintain antibiotic resistance.   

Minor comments:

Line 54, remove period after ‘microbiota’

Line 59-61, need to reference this statement and the average of < 1%.

Line 64-66, “The proportion of these organisms (also Enterococcus) in the microbiota increases in aged and hospitalized individuals and in people from low-income countries with inadequate sanitation”………….Reference needed.

Author Response

REVIEWER 3

A!. The authors present a review of potential physiological stresses and propose to utilize these stresses and test the hypothesis that such stresses may affect the cost-benefit balance of drug-resistant opportunistic pathogens in the gut. This is a very important topic and will be of interest to a wide audience in the scientific community. The authors did an excellent job summarizing those physiological stresses and enhance their report with some schematics.

Q1. We are most grateful for your kind words supporting our hypothesis.

A2. The authors do not address the issue of resistance gene transfer between non-pathogenic commensals and pathogens. As an example, the vanB-type vancomycin resistance transposon, which is commonly carried by anaerobic gut commensals of the phylum Firmicutes [Stinear TP, Olden DC, Johnson PD, Davies JK, Grayson ML. Enterococcal vanB resistance locus in anaerobic bacteria in human faeces. Lancet. 2001. Mar;357(9259):855–6. 10.1016/S0140-6736(00)04206-9]. Another issue is the difficulty of testing specific immunological stresses on gut microbiota or in the testing of the proposal provided by the authors. The review could also benefit the reader by considering the case of drug-resistant C. difficile. A real problem to address and a pathogen can apparently withstand the physiological stresses in the gut and is able to persist in the host and maintain antibiotic resistance.   

Q2. Thanks to the reviewer for his insightful comments. We added a sentence concerning the “question of resistant commensals” at the end of our manuscript, and inserted the suggested reference. We do not have entered in the detail of C. difficile, as we understand this case is included in our statement concerning “commensal organisms populations expressing and spreading resistance, including anaerobes”.

Minor comments:

Q3. Line 54, remove period after ‘microbiota’

A3. Done.

Q4. Line 59-61, need to reference this statement and the average of < 1%.

A4. References have been added in this paragraph.

Q5. Line 64-66, “The proportion of these organisms (also Enterococcus) in the microbiota increases in aged and hospitalized individuals and in people from low-income countries with inadequate sanitation”………….Reference needed

A5. References have been added after such statement.

Reviewer 4 Report

Comments and Suggestions for Authors

In the manuscript by Baquero et al., the authors discuss the possible impact of stress on the antibiotic resistance of gut bacteria. The impact of several stress factors on the fitness of bacterial cells is discussed and strategies how to determine this are suggested.

It is hypothesized in the abstract that “ effect of combinations of fitness costs should be tested in antibiotic resistant bacteria with susceptible ones as control”. This is an opinion, not a hypothesis. Please change accordingly.

From page 3 till 8 the impact of different kinds of stress, like acid, bile and antibiotics, is discussed. Each stress component has its own section. It is a long part of text to read and coherence between the different stress factors is missing.

Page 9, first sentence: this is a long sentence and it is not clear to which “both of fitness costs” the authors are referring to.

It is a complicated manuscript to read. In general, the manuscript contains a lot of long sentences. Regularly a sentence has to be read twice in order to understand what is actually mentioned in the sentence. I would like to recommend to use shorter sentences which makes it easier to read the manuscript.

Comments on the Quality of English Language

Difficult to read due to the use of long sentences.

Author Response

Reviewer 4

Q1. In the manuscript by Baquero et al., the authors discuss the possible impact of stress on the antibiotic resistance of gut bacteria. The impact of several stress factors on the fitness of bacterial cells is discussed and strategies how to determine this are suggested. It is hypothesized in the abstract that “ effect of combinations of fitness costs should be tested in antibiotic resistant bacteria with susceptible ones as control”. This is an opinion, not a hypothesis. Please change accordingly.

A1. You are absolutely right. This is an opinion, not a hypothesis. In fact is an opinion that gives rise to the possible formulation of a hypothesis.  In fact there are pieces of evidence supporting our opinion (that is the essential of our work), that is why I deviated to the “hypothesis”  wording. We have corrected this mistake.

Q2. From page 3 till 8 the impact of different kinds of stress, like acid, bile and antibiotics, is discussed. Each stress component has its own section. It is a long part of text to read and coherence between the different stress factors is missing.

A2. It is difficult to establish a more coherent presentation of the different, heterogeneous factors involved in the different types of stress. In addition, the lack of experimental information about stresses in antibiotic-resistant and -susceptible bacteria and the possible effects of combinations of stressors in different parts of the gut makes unreliable (at the moment) to enumerate the stressors in a better ordered way. We did the best we could.

Q3. Page 9, first sentence: this is a long sentence and it is not clear to which “both of fitness costs” the authors are referring to.

A3. We splitted this long sentence and now is more clear. We refer here to “fitness costs” in general.

Q4. It is a complicated manuscript to read. In general, the manuscript contains a lot of long sentences. Regularly a sentence has to be read twice in order to understand what is actually mentioned in the sentence. I would like to recommend to use shorter sentences which makes it easier to read the manuscript.

A4. The topic is complex and necessarily has an heterogeneous content, based on different pieces of published evidence originated in different scenarios. We rephrased a number of sentences, but some of them are difficult to split as (for us) they have a meaning only if the conceptual pieces are assembled. In any case the manuscript has ben thoughtfully reviewed by two highly professional native English experts.